# Assembly and Comparative Analysis of the Complete Mitochondrial Genome of Two Species of Calla Lilies (*Zantedeschia*, Araceae)

**DOI:** 10.3390/ijms24119566

**Published:** 2023-05-31

**Authors:** Yanbing Guo, Ziwei Li, Shoulin Jin, Shuying Chen, Fei Li, Hongzhi Wu

**Affiliations:** 1College of Animal Science and Technology, Yunnan Agricultural University, Kunming 650201, China; yanbingguo@139.com (Y.G.); diwuzisang@163.com (Z.L.); 2College of Horticulture and Landscape, Yunnan Agricultural University, Kunming 650201, China; 13708477248@163.com; 3College of Agriculture and Biotechnology, Yunnan Agricultural University, Kunming 650201, China; jslwhz@163.com (S.J.);

**Keywords:** *Z. aethiopica*, *Z. odorata*, hybrid assembly, mitochondrial genome, comparative analysis, maternal mitochondrial inheritance

## Abstract

In this study, the mitochondrial genomes of two calla species, *Zantedeschia aethiopica* Spreng. and *Zantedeschia odorata* Perry., were assembled and compared for the first time. The *Z. aethiopica* mt genome was assembled into a single circular chromosome, measuring 675,575 bp in length with a 45.85% GC content. In contrast, the *Z. odorata* mt genome consisted of bicyclic chromosomes (chromosomes 1 and 2), measuring 719,764 bp and exhibiting a 45.79% GC content. Both mitogenomes harbored similar gene compositions, with 56 and 58 genes identified in *Z. aethiopica* and *Z. odorata*, respectively. Analyses of codon usage, sequence repeats, gene migration from chloroplast to mitochondrial, and RNA editing were conducted for both *Z. aethiopica* and *Z. odorata* mt genomes. Phylogenetic examination based on the mt genomes of these two species and 30 other taxa provided insights into their evolutionary relationships. Additionally, the core genes in the gynoecium, stamens, and mature pollen grains of the *Z. aethiopica* mt genome were investigated, which revealed maternal mitochondrial inheritance in this species. In summary, this study offers valuable genomic resources for future research on mitogenome evolution and the molecular breeding of calla lily.

## 1. Introduction

The Araceae family, one of the largest monocot families, comprises approximately 3667 species across 143 genera [1]. Many species within the family hold global significance in medicine and horticulture and as edible plants. The representative genus belonging to the Araceae family, the *Zantedeschia* genus, are herbaceous monocotyledonous ornamental geophytes indigenous to the swampy or mountainous regions of South Africa [2]. The genus encompasses eight species, divided into two sections. The *Zantedeschia* section includes two species, *Z. aethiopica* and *Z. odorata*, characterized by a white spathe and rhizomatous storage organs. In contrast, the *Aestivea* section comprises *Z. albomaculata*, *Z. elliottiana*, *Z. jucunda*, *Z. pentlandii*, *Z. rehmannii*, and *Z. valida* with various spathe colors and tuberous storage organs [2,3]. The colored species of *Zantedeschia* are of particular ornamental value due to the prolonged flowering period and vivid flower spathes obtained as a result of the interspecific crossing of *Z. rehmannii*, *Z. elliottiana*, *Z. pentlandii*, and *Z. albomaculata* [3,4]. On the contrary, *Z. aethiopica* has not only decorative but also practical applications in traditional medicine in some regions of Africa [5].

Prior research has indicated that colored *Zantedeschia* species are highly susceptible to *Pectobacterium* bacteria, causing considerable economic losses, whereas *Z. aethiopica* demonstrates greater tolerance to various abiotic and biotic stressors [6]. Nonetheless, interspecific hybrids between them experience plastome–genome incompatibility (PGI), leading to the production of albino offspring due to endosperm degeneration and chlorophyll-deficient abnormal embryos [7,8]. Consequently, the introduction of desirable traits, such as resistance (from *Z. aethiopica*) and showy spathes (from colored *Zantedeschia*), into hybrid offspring through conventional breeding strategies remains challenging.

Mitochondrial (mt) and chloroplast (cp) genomes possess semi-autonomous genetic systems in higher plant cells and carry essential genetic information. Previous studies have elucidated the PGI mechanism in *Zantedeschia* and revealed that *Z. aethiopica* exhibits biparental plastid inheritance and maternal mitochondrial inheritance, as evidenced by restriction fragment length polymorphism and cytological analyses [8,9]. However, a comprehensive understanding of the underlying molecular mechanisms remains elusive, primarily due to the scarcity of essential genetic resources, particularly the absence of a reported mitochondrial genome for *Zantedeschia*. This deficiency severely impedes further investigation. Therefore, studying the *Zantedeschia* mitochondrial genome could help decipher the molecular mechanisms behind hybrid sterility and contribute to the improvement of *Zantedeschia* varieties. As of 23 March 2023, the NCBI database has published 10,385 chloroplast genomes, 1296 plastomes, and a mere 596 plant mitochondrial genomes (https://www.ncbi.nlm.nih.gov/genome/browse/#!/organelles/; accessed on 23 March 2023). This disparity underscores the complexity and difficulty of assembling plant mitogenomes [10]. Current evidence reveals that plant mt genomes exist in various structures, including circular molecules, linear conformations, branched structures, and polycyclic molecules [11,12,13,14,15]. Although plant mitogenomes differ significantly in size (from 66 kb to 11.3 Mb) [16,17], their gene contents remain highly conserved, with 24 core protein-coding genes (PCGs) in angiosperm mitogenomes [18]. Due to the complexity of studying plant mitogenomes, currently, only two Araceae species’ mitogenomes have been assembled and deposited in the NCBI database: *Spirodela polyrhiza* (NC_017840) and *Amorphophallus albus* (NC_066968). Mitogenome sequences still remain unavailable for most species of the family, impeding the development of new varieties with broader adaptive and commercial traits.

In this study, two complete mt genomes of the *Zantedeschia* species (*Z. aethiopica* and *Z. odorata*) were sequenced and annotated. synonymous codon usage (RSCU), sequence repeats, foreign DNA fragments, and RNA editing sites were analyzed in the two calla species. Available respective data on the other Araceae was also used for comparison with the aim of better understanding the evolution and origin of the *Zantedeschia* mitogenome. In addition, the expression of 24 core genes was determined in the gynoecium, stamens, and pollen grains of *Z. aethiopica* using RT-PCR. The results presented here provide a unique insight into the mitochondrial evolution of calla species, and it is also a basis for a depicted PGI mechanism and the molecular design breeding of *Zantedeschia* spp.

## 2. Results

### 2.1. Features of the Zantedeschia Mitogenome

The *Z. aethiopica* circular mt genome was 675,575 bp in length. The GC content was 45.85% (Table 1). For the *Z. odorata* mt genome, however, two circular contigs (chromosomes 1 and 2) were obtained, and it was 719,764 bp long with an overall GC content of 45.79%. The lengths of circular chromosomes 1 and 2 were 432,076 bp and 287,688 bp with GC contents of 45.95% and 45.54% (Table 1), respectively. The functional categorization and physical locations of the annotated genes are presented in Figure 1A,B. A total of 34 and 36 protein-coding genes (PCGs) were annotated in *Z. aethiopica* (the *rpl2* and *rps10* genes were absent) and *Z. odorata*, respectively. In addition, both species shared the same rRNA genes (3) and tRNA genes (19). These results indicated that the gene content of the two mt genomes was relatively conserved. Most PCGs occurred in single copies within the two mitogenomes (Table 2). However, *ccmC* presented two gene copies in *Z. odorata*. The two mitogenomes also contained the same RNA genes (*rrn5*, *rrn18*, and *rrn26*) and tRNAs (*trnA*-*UGC*, *trnC*-*GCA*, *trnD*-*GUC*, *trnE*-*UUC*, *trnF*-*GAA*, *trnfM*-*CAU*, *trnG*-*GCC*, *trnH*-*GUG*, *trnI*-*CAU*, *trnK*-*UUU*, *trnM*-*CAU*, *trnN*-*GUU*, trnP-*UGG*, *trnQ*-*UUG*, *trnS*-*GCU*, *trnS*-*UGA*, *trnV*-*GAC*, *trnW*-*CCA*, and *trnY*-*GUA*). It is worth noting that although most of the 19 tRNA genes are single-copy genes, the *Z. aethiopica* mitogenome possessed two copies of the *trnH*-*GUG*, *trnI*-*CAU*, *trnM*-*CAU*, and *trnN*-*GUU* genes, whereas the *Z. odorata* mitogenome possessed two copies of the *trnH*-*GUG*, *trnI*-*CAU*, *and trnM*-*CAU* genes.

### 2.2. Codon Usage Analysis of the PCGs

The codon usage analysis of 34 and 36 PCGs in the mt genomes of *Z. aethiopica* and *Z. odorata* was performed, respectively, with the codon usage per amino acid displayed in Appendix A. Relative synonymous codon usage (RSCU) values greater than 1 are considered to indicate amino acid preferences. As depicted in Figure 2, in addition to the RSCU values of both the initial codon AUG (Met) and UGG (Trp) being 1, there was also a general preference for codon usage in mitochondrial PCGs. In *Z. aethiopica*, GCU (Ala) had the highest RSCU value of 1.60 for the GCU codon in mitochondrial PCGs. Moreover, the stop codon UAA exhibited codon usage preference with an RSCU value of 1.59. For *Z. odorata*, however, the stop codon UAA demonstrated a high degree of codon usage preference with the highest RSCU value among the mitochondrial PCGs, with this value reaching 1.76. This was followed by alanine (Ala) with GCU preferred with an RSCU value of 1.61. Notably, the maximum RSCU values for lysine (Lys) and phenylalanine (Phe) were less than 1.2 within two calla species, indicating no strong codon usage preference. 

### 2.3. Repeats Analysis of Zantedeschia Mitogenomes

Simple sequence repeats (SSRs) in the mt genomes of *Z. aethiopica* and *Z. odorata* were identified using MISA software (v2.1) (https://webblast.ipk-gatersleben.de/misa/; accessed on 1 March 2023). In the *Z. aethiopica* mt genome, 139 SSRs were found, with monomeric and dimeric SSRs occupying 31.65% of the total SSRs (Figure 3A). Among the 13 monomer SSRs, adenine (A) monomer repeats constituted 53.85% (7), while the most frequent dimeric SSR was the TA repeat, comprising 29.03%. Tetrameric SSRs were the most common, representing 51.08% (71) of the total SSRs (Appendix A). For the *Z. odorata* mt genome, 79 and 66 SSRs were detected in chromosomes 1 and 2, respectively, and the monomeric and dimeric SSRs occupied 31.65% and 36.36% of the respective total SSR. In chromosome 1, adenine (A) monomer repeats comprised 80.00% (4) of the five monomer SSRs, and TA repeats were the most frequent dimeric SSRs, making up 30.00%. The most common SSRs were tetranucleotide, representing 45.57% of the SSRs. In chromosome 2, adenine (A) monomer repeats constituted 62.50% (5) of the eight monomer SSRs, and CT repeats were the most frequent dimeric SSRs, comprising 31.25%. Tetranucleotide SSRs were also the most common, accounting for 50.00% of the SSRs (Appendix A).

Tandem repeats, also called satellite DNA, consist of core repeating units of approximately 1 to 200 bases repeated several times in tandem. These repeats are widely present in eukaryotic genomes and some prokaryotes. As illustrated in Figure 3B, 47 tandem repeats ranging from 11 to 66 bp in length with a matching degree greater than 66% were found in the *Z. aethiopica* mt genome (Appendix A). In the *Z. odorata* mt genome, 32 tandem repeats ranging from 11 to 66 bp with a matching degree greater than 66% were identified in chromosome 1. In chromosome 2, 25 tandem repeats ranging from 14 to 56 bp with a matching degree greater than 73% were detected (Appendix A).

Dispersed repeats in the mt genomes of Z. aethiopica and Z. odorata were also analyzed (Appendix A). In *Z. aethiopica*, a total of 1746 pairs of dispersed repeats with lengths of 30 or more were observed, including 891 pairs of palindromic repeats (up to 189 bp) and 855 pairs of forward repeats (up to 283 bp). In chromosome 1 of *Z. odorata*, 1746 pairs of dispersed repeats with lengths of 30 or more were identified, including 891 pairs of palindromic repeats (the longest being 210 bp) and 855 pairs of forward repeats (the longest being 164 bp). In chromosome 2 of *Z. odorata*, 475 pairs of dispersed repeats with lengths of 30 or more were observed, including 193 pairs of palindromic repeats (the longest being 106 bp) (Appendix A). No complement and reverse repeats were detected within the two calla species.

### 2.4. Analysis of Homologous Fragments of Mitochondria and Chloroplasts

The sequence similarity analysis revealed a total of 29 fragments homologous to the mt and cp genome in *Z. aethiopica*, with a combined length of 48,353 bp, constituting 7.16% of the total mitogenome length (Figure 4A, Appendix A). This homologous sequence is called mitochondrial plastid sequences (MTPTs). The longest fragment among these, MTPT15, spans 4760 bp. Annotation of these homologous sequences identified 12 complete genes, including eight (PCGs) (*atpB*, *atpE*, *ndhK*, *ndhC*, *psaA*, *rpl23*, *rpoB*, and *rps7*) and four tRNA genes (*trnI*-*CAU*, *trnN*-*GUU*, *trnM*-*CAU*, *trnW*-*CCA*). In *Z. odorata*, 25 homologous fragments were identified, with a total length of 47,946 bp, representing 6.66% of the mt genome (Figure 4B, Appendix A). The longest fragment, MTPT7, measured 4333 bp. These homologous fragments encompassed 12 annotated genes, including nine PCGs (*atpB*, *atpE*, *ndhK*, *ndhC*, *psaA*, *rpl2*, *rpl23*, *rpoB*, and *rps7*) and three tRNA genes (*trnI*-*CAU*, *trnM*-*CAU*, and *trnW*-*CCA*). 

### 2.5. Prediction of RNA Editing Sites in PCGs

The PREP (Predictive RNA Editors for Plants) suite was employed to identify potential RNA editing sites within the PCGs of the *Z. aethiopica* and *Z. odorata* mt genomes. As shown in Figure 5, the two mitogenomes exhibited similar editing site numbers. For example, *Z. aethiopica* and *Z. odorata* contained 522 and 512 C-U editing sites, respectively. Approximately 325 RNA editing sites occurred at the second base of each codon in two species. In contrast, approximately 187 sites occurred at the first base of the codon (Appendix A). For *Z. aethiopica*, the *CcmB* gene exhibited the highest number of editing sites among all mitochondrial genes, with 41 RNA editing sites. The next highest number was 39 for *ccmFN*. The rps14 gene was predicted to have only one RNA editing event, the lowest among all genes. Interestingly, the types and numbers of RNA editing events in the *Z. odorata* mt genome were consistent with those in *Z. aethiopica*, except for the *ccmC*, *cox2*, *rpl2*, and *rpl5* genes.

### 2.6. Phylogenetic and Synteny Analysis

To ascertain the evolutionary status of *Z. aethiopica* and *Z. odorata*, phylogenetic trees were constructed based on 18 conserved mitochondrial PCGs (*atp1*, *atp4*, *atp6*, *ccmB*, *ccmC*, *ccmFC*, *ccmFN*, *cob*, *cox1*, *cox2*, *matR*, *nad1*, *nad2*, *nad4*, *nad5*, *nad6*, *nad7*, and *nad9*) from five orders of angiosperms (Poales, Arecales, Asparagales, Alismatales, and Ranunculales). *Pulsatilla dahurica* and *Aconitum kusnezoffii* of the Ranunculales were designated as outgroups. As depicted in Figure 6, the phylogenetic topology based on mitochondrial DNA was consistent with the latest APG (Angiosperm Phylogenetic Group) classification (Figure 7), with the phylogenetic tree providing robust support for the grouping of *Z. aethiopica* and *Z. odorata*, which are closely related to *S. polyrhiza* of the Araceae family within Alismatales.

As illustrated in Figure 8, the dot plot analysis identified a positive repeat sequence of approximately 100 kb in length shared between the *Z. aethiopica* and *Z. odorata* mitogenomes. Nearly 20 kb collinearity blocks were identified in the dot plot comparing *Z. aethiopica* and *S. polyrhiza*, and similarly sized collinearity blocks were also detected in the *Z. odorata* and *S. polyrhiza* mt genomes. These findings indicate that the arrangement order of collinear blocks varies between individual mitochondrial genomes, suggesting that the *Z. aethiopica* and *Z. odorata* mt genomes have undergone extensive genome rearrangements with nearby species and that the mt genomes are highly divergent in structure.

### 2.7. RT-PCR Validation of Mitogenome Core Genes (24) of in Z. aethiopica

A previous study has demonstrated that the mitochondrial DNA of mature pollen grains in *Z. aethiopica* is significantly degraded, as evidenced by the cytological findings, which suggests maternal mitochondrial inheritance [9]. Here, the expression of 24 core genes was tested in the gynoecium, stamens, and mature pollen grains of *Z. aethiopica* using RT-PCR (Figure 9). As depicted in Figure 9D, 11 genes (*atp1*, *atp4*, *atp8*, *atp9*, *nad1*, *nad4L*, *nad5*, *nad6*, *nad7*, *ccm*, and *ccmC*) were successfully amplified in both the gynoecium and stamens, while only four genes (*atp4*, *atp8*, *atp9*, *and nad7*) were detected in mature pollen grains. Furthermore, the detected genes displayed degradation in mature pollen grains, with the exception of the *atp9* gene. These results provide molecular evidence supporting the notion of maternal mitochondrial inheritance in *Z. aethiopica*.

## 3. Discussion

### 3.1. The Characteristics of Zantedeschia Mitogenome

The plant mt genome is an evolutionarily dynamic entity characterized by remarkable diversity in genome size, genome structure, and gene content [19]. Higher plant mitogenomes exhibit considerable variation in size and gene content, ranging from 66 Kb to 11.3 Mb and encompassing 19 to 64 known genes [16,17]. Plant mitogenomes display diverse conformations, with the majority of the assembled mitogenomes being circular. However, other structures, such as polycyclic [15,20,21], linear [13], and multi-branched structures [14], also exist. In this study, a hybrid assembly strategy was employed, combining Illumina short-read and Oxford Nanopore long-read sequencing to complete the mitogenomes of *Z. aethiopica* and *Z. odorata*, members of the *Zantedeschia* genus in the Araceae family. The comparative analysis revealed conservation of mt genome size and gene contents within the *Zantedeschia* genus, albeit with complex genome structures. The *Z. aethiopica* mt genome was a single-ring structure of 675,575 bp, comprising 56 genes, while the *Z. odorata* mt genome iwa assembled into two circular chromosomes totaling 719,764 bp and including 58 genes (Figure 1, Table 1 and Table 2).

Plant mt genomes are abundant in repeat sequences, which consist of SSRs, tandem repeats, short repeats, and large repeats [12,22]. It has been reported that repeat sequences play a crucial role in shaping the mt genome [22,23]. In this study, SSRs, tandem repeats, and dispersed repeats were investigated in the two sequenced *Zantedeschia* mt genomes. The comparative repeat analyses of the two *Zantedeschia* mitogenomes (Figure 3) revealed an absence of repeats larger than 350 bp, which might contribute to the stability of genome sizes and gene contents within the *Zantedeschia* genus. While the highly complex repeat patterns may be responsible for the intricate mt genome structure in plant species, the mitogenome structure is not solely determined by repeats, which might explain the distinct structures of the two *Zantedeschia* mt genomes.

Plant multi-chromosomal mitogenomes often exhibit an uneven distribution of functional genes between chromosomes, with numerous chromosomes in plant multi-chromosomal mitogenomes lacking functional genes [24]. In this study, it was observed that 24 PCGs were unevenly distributed between chromosome 1 (17 PCGs) and chromosome 2 (7 PCGs) in the *Z. odorata* mt genome. Further transcriptome studies are needed to determine whether these genes can be appropriately expressed in plant mitochondria.

### 3.2. RNA Editing and Intracellular Gene Transfer of Zantedeschia Organelle Genomes

RNA editing is a post-transcriptional process that occurs in all higher plants [25], contributing to the maintenance of amino acid sequence conservation in essential functional proteins of the mt genome [26,27]. RNA editing enhances the diversity of gene products, leading to the translation of novel proteins with functional roles [28,29,30]. Previous studies have reported approximately 441 RNA-editing sites within 36 genes in Arabidopsis and 491 RNA-editing sites within 34 genes in rice [31,32]. In this study, it was observed that the number of RNA editing sites in PCGs was remarkably conserved in *Z. aethiopica* and *Z. odorata*, with 522 RNA editing sites within 34 PCGs in *Z. aethiopica* and 512 RNA editing sites within 36 PCGs in *Z. odorata*; both of these results demonstrate C-U RNA editing.

In angiosperm plants, gene transfer from cp to mt genomes is a common occurrence during long-term evolution [33,34]. In this study, 29 MTPTs were identified in *Z. aethiopica* and 25 MTPTs in *Z. odorata*. These MTPTs spanned 48,353 bp and 47,946 bp in length, accounting for 7.16% and 6.66% of the *Z. aethiopica* and *Z. odorata* mitogenomes, respectively. These results support the conclusion that approximately 1–10.3% of mtDNA is derived from the chloroplast genome [35].

### 3.3. Phylogenetic Inference and Synteny Analysis

In the present study, the phylogenetic relationship of *Z. aethiopica* and *Z. odorata* was analyzed with representative taxa based on mt genome information. The resulting phylogenetic trees reflected well-defined taxonomic relationships between taxa. The mt genome sequences of *Z. aethiopica* and *Z. odorata* have good co-linear blocks, indicating that the two species have undergone extensive rearrangement phenomena. It can be speculated that a broken fusion occurred between their different chromosomes, i.e., that *Z. aethiopica* and *Z. odorata* did not differentiate at the same time, but evolved from one species to another.

### 3.4. Z. aethiopica Displays Maternal Mitochondrial Inheritance

Maternal inheritance of mitochondria has been reported in a group of angiosperms, including *Actinidia deliciosa* [36], *Medicago sativa* [37], *Turnera ulmifolia* [9], and *Z. aethiopica* [9], based on cytological evidence. In this study, molecular evidence was obtained for *Z. aethiopica*, revealing that most core genes were significantly degraded in mature pollen grains, as demonstrated by RT-PCR (Figure 9). Only 4 out of 24 core genes were normally amplified, with the others displaying various levels of degradation and the majority not being detected. These results are consistent with Ji [9], which also found that most mitochondrial DNA was significantly degraded in *Z. aethiopica* mature pollen grains. The mechanism of maternal mitochondrial inheritance warrants further investigation in the future.

### 3.5. The Prospects of Zantedeschia Mitogenome Research

Future studies could analyze the structure of the calla lily mitochondrial genome at the population level, including parental and hybrid offspring. Cryoelectron microscopy could be employed to visualize the actual structural characterization of the calla lily mitochondrial genome. Additionally, hybridization experiments should be conducted to obtain multiple interspecific hybrid offspring to investigate the relationship between PGI and the mt genome. These studies will provide a substantial foundation for understanding mt genome evolution and facilitate mitochondrial-based breeding.

## 4. Materials and Methods

### 4.1. Plant Materials, DNA Extraction, and Sequencing

Fresh young leaves of *Z. aethiopica* and *Z. odorata* were collected from the Garden Greenhouse (Geospatial coordinates: 102.83945, 24.88627), College of Landscape and Horticulture, Yunnan Agricultural University (Kunming, Yunnan Province, China). High-quality genomic DNA was extracted using the CTAB method [38]. DNA sample quality was evaluated with agarose gel electrophoresis, and the concentration was measured using a Nanodrop instrument (Thermo Fisher Scientific, Waltham, MA, USA, 2000c UV-Vis). The qualified samples were sent to Wuhan GrandOmics Technology Co., Ltd. (http://www.grandomics.com; accessed on 12 December 2022) for Illumina sequencing and Oxford Nanopore sequencing.

### 4.2. Assembly and Annotation of Mitochondrial Genomes

Flye (v.2.9.1-b1780) [39] was used to de novo assemble the long-reads of the two *Zantedeschia* species with the parameters of ‘-min-overlap 4000’. For the assembled contigs, the draft mitogenome based on Nanopore long-reads was identified using the BLASTn [40] program. Here specifically, makeblastdb was used to construct a database for the assembled sequences by Flye, and then the conserved mitochondrial genes from *Liriodendron tulipifera* (NC_021152.1) were used as the query sequence to identify contigs containing conserved mitochondrial genes. The short-reads and long-reads were then mapped to these contigs, and all mapped reads were retained using BWA and SAMTools [41,42]. Finally, Illumina short-reads and Nanopore long-reads were combined for hybrid assembly by using Unicycler (v0.5.0) [43], with the parameters of ‘-kmers 27, 53, 71, 87, 99, 111, 119, 127’. In this step, the mapped Illumina short-reads were first assembled by calling spades [44], and then, the Nanopore long-reads were used to resolve the regions of repetitive sequences of the assembly by calling minimap2 [45]. The GFA format files generated by Unicycler were visualized using Bandage [46]. Unicycler generated one circular contigs for *Z. aethiopica* and two circular contigs for *Z. odorata*.

The protein and rRNA genes of the mt genome were annotated using the online GeSeq tool [47] and local BLASTN [40], with *Spirodela polyrhiza* (NC_017840), *Arabidopsis thaliana* (NC_037304), and *Liriodendron tulipifera* (NC_021152.1) as references, and the tRNAs were identified with tRNAscan (version 1.4) [48]. Apollo software (v1.11.8) was used to manually modify and correct annotation errors for each mitochondrial genome [49]. The mitochondrial genome maps were drawn using Organellar Genome DRAW (OGDRAW) (v1.3.1) [50].

### 4.3. Analysis of Codon Usage and Repeated Sequences

The protein-coding sequences of the genome were extracted using Phylosuite (v1.2.2) [51]. Mega 7.0 [52] was used to perform codon preference analysis on the PCGs of the mitochondrial genome and calculate RSCU values. The MISA [53], TRF (https://tandem.bu.edu/trf/trf.unix.help.html; accessed on 1 March 2023), and REPuter web server (https://bibiserv.cebitec.uni-bielefeld.de/reputer/; accessed on 1 March 2023) were used to identify repetitive sequences including microsatellite sequence repeats, tandem repeats, and dispersed repeats [54,55]. The results were visualized using Excel (2021) software.

### 4.4. Analyses of Chloroplast to Mitochondrion DNA Transformation and RNA Editing

The cp genome of *Z. aethiopica* (MH743155) and *Z. odorata* (MG432242) were downloaded from the NCBI Organelle Genome Resources Database. The homologous fragments in the mt and cp genomes were identified using BLASTN [40] software (v0.0.5), and the results were visualized using the Circos package (v0.69) [56]. Putative RNA editing sites in the PCGs of two calla species were predicted at a cutoff value = 0.2 criterion using the web server (http://www.prepact.de/; accessed on 2 March 2023) PREPACT3 (v3.12.0) [57].

### 4.5. Analysis of Phylogeny and Synteny

The mitogenomes of closely related species were selected and downloaded (https://www.ncbi.nlm.nih.gov; accessed on 4 March 2023) based on their affinity. A total of 30 mitogenomes were downloaded, including two outgroups (*Pulsatilla dahurica*: NC_071219.1 and *Aconitum kusnezoffii*: NC_053920.1). These species (Appendix A) were then used for phylogenetic analysis along with the two *Zantedeschia* species that were newly sequenced. Firstly, the 24 core orthologous genes among the analyzed species were extracted by using PhyloSuite (v.1.2.2) [51]. Then, the corresponding nucleotide sequences were aligned using MAFFT (v7.471) [58]. Next, these aligned sequences were concatenated and used to construct the phylogenetic trees. The maximum likelihood (ML) method was implemented in IQ-TREE (version 2.1.4-beta) [59] under the GTR + F + I + I + R2 model. The bootstrap analysis was performed with 1000 replicates. The phylogenetic tree was edited on the online website ITOL [60].

Four closed species (*Spirodela polyrhiza* (NC_017840.1), *Butomus umbellatus* (NC_021399.1), *Zostera japonica* (NC_068803.1), and *Stratiotes aloides* (NC_035317.1)) were selected for the co-linear analysis with *Z. aethiopica* and *Z. odorata*. Conserved homologous sequences, referred to as collinear blocks, were identified using the BLASTn program [39] with parameters of ‘-evalue 1e-5, -word_size 9, -gapopen 5, -gapextend 2, -reward 2, and -penalty 3’. Only collinear blocks that exceeded 500 bp in length were chosen for further analysis. MCscanX [61] was utilized to generate the multiple synteny plot based on the results obtained from the BLASTn program. Pairwise comparisons (*Z. aethiopica*, *Z. odorata*, and *S. polyrhiza*) of dot plots were generated, and conserved co-linear blocks were plotted using MAFFT with default parameters [58].

### 4.6. RT-PCR of Mitogenome Core Genes of Gynoecium, Stamens, and Mature Pollen Grains in Z. aethiopica

*Z. aethiopica* gynoecium, stamens, and mature pollen grains were collected from the Garden Greenhouse (Geospatial coordinates: 102.83945, 24.88627), College of Landscape and Horticulture, Yunnan Agricultural University (Kunming, Yunnan Province, China). Total RNA was isolated from the gynoecium, stamens, and pollen grains using an RNA extraction kit (TIANGEN, DP432, Beijing, China). First-strand cDNA was synthesized using a FastKing RT Kit (TIANGEN, KR116, Beijing, China). RT-PCR was performed using the primers listed in Appendix A, with the following amplification conditions: 94 °C for 2 min; 32 cycles of 94 °C for 30 s, 56 °C for 30 s, and 72 °C for 2 min; and a final extension at 72 °C for 10 min.

## 5. Conclusions

In this study, two mt genomes of calla lily (*Z. aethiopica* and *Z. odorata*) were sequenced, assembled, and annotated, and comprehensive analyses were conducted based on the DNA and amino acid sequences of the annotated genes. The two calla species’ mitogenomes shared similar gene contents but varied in structure. The *Z. aethiopica* mt genome’s length of the one circular chromosome was 675,575 bp, and the GC content was 45.85%. Meanwhile, the *Z. odorata* mt genome consisted of two circular chromosomes with a total genome size of 719,764 bp and a GC content of 45.79%. By comparing the mt genomes and the cp genomes, 29 and 25 homologous fragments were found in the two species, respectively. Furthermore, codon usage, repeated sequences, RNA editing, and synteny were analyzed. Moreover, phylogenetic trees based on the mitogenomes of 30 species contributed to the taxonomic classification of *Z. aethiopica* and *Z. odorata*. Lastly, maternal mitochondrial inheritance was demonstrated in *Z. aethiopica* using RT-PCR. This study provides pivotal genomic resources for understanding and utilizing this vital ornamental plant in the future, particularly in elucidating the PGI mechanism in plants.

## Figures and Tables

**Figure 1 ijms-24-09566-f001:**
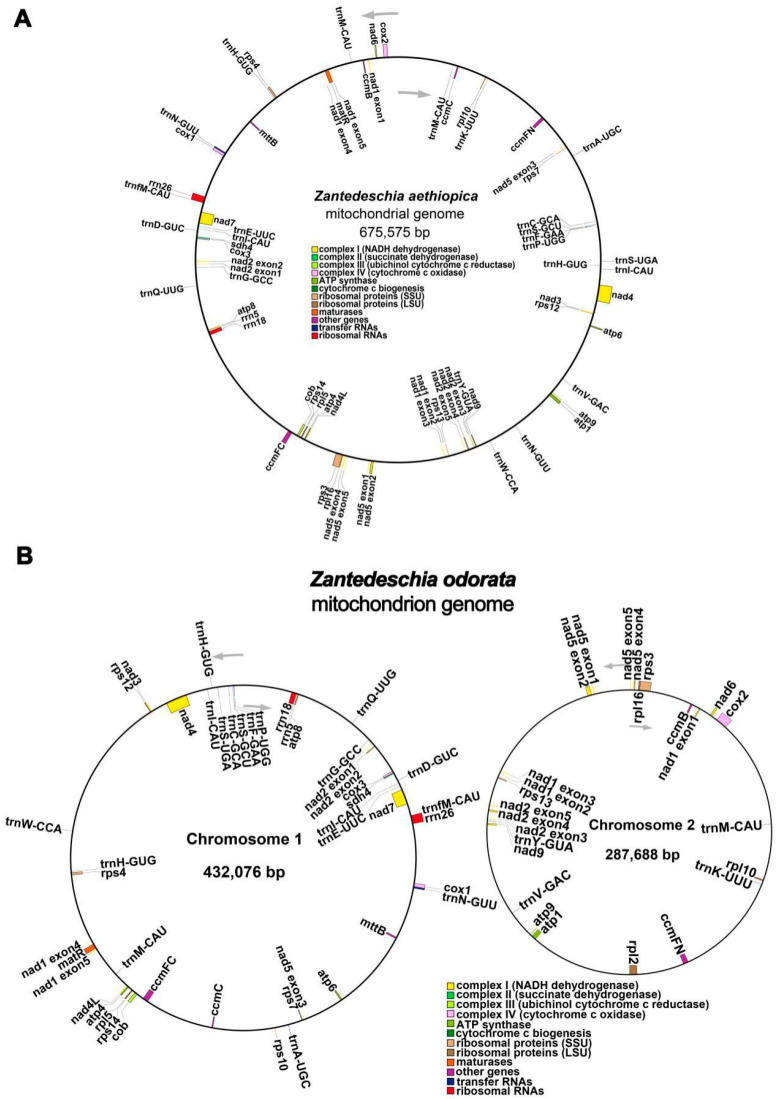
Circular maps of the *Z. aethiopica* (**A**) and *Z. odorata* (**B**) mt genomes. Genes shown on the outside and inside of the circle are transcribed clockwise and counterclockwise, respectively. Genes belonging to different functional groups are color-coded.

**Figure 2 ijms-24-09566-f002:**
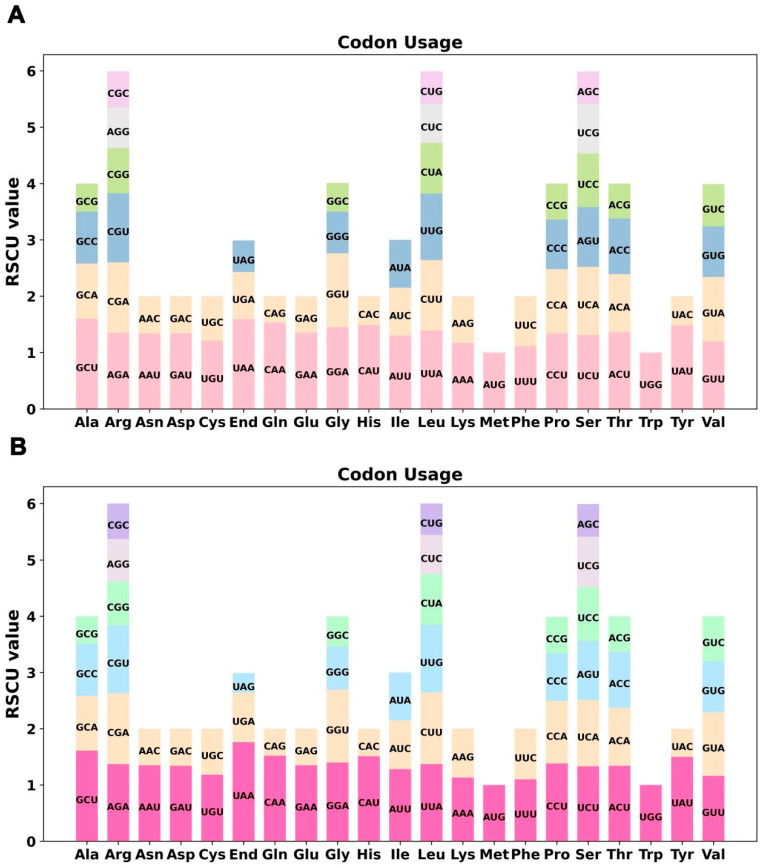
*Z. aethiopica* (**A**) and *Z. odorata* (**B**) mitogenome relative synonymous codon usage. The codon families are shown on the X-axis. The RSCU values are the number of times a particular codon is observed relative to the number of times that codon would be expected for uniform synonymous codon usage.

**Figure 3 ijms-24-09566-f003:**
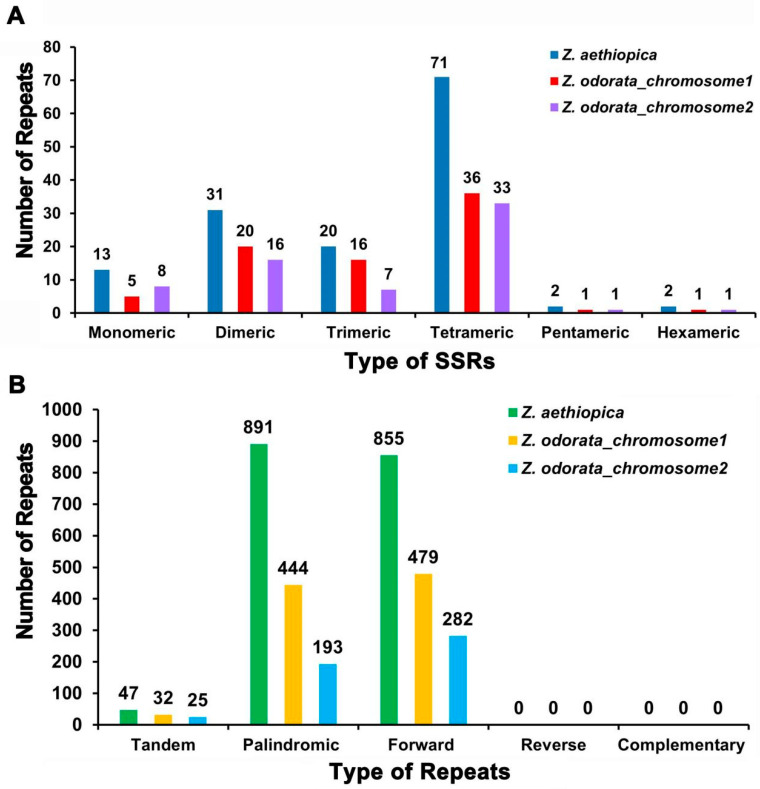
Detected repeats in the *Z. aethiopica* and *Z. odorata* mitogenomes. (**A**) Type and number of detected SSRs repeats. (**B**) Type and number of detected tandem and dispersed repeats.

**Figure 4 ijms-24-09566-f004:**
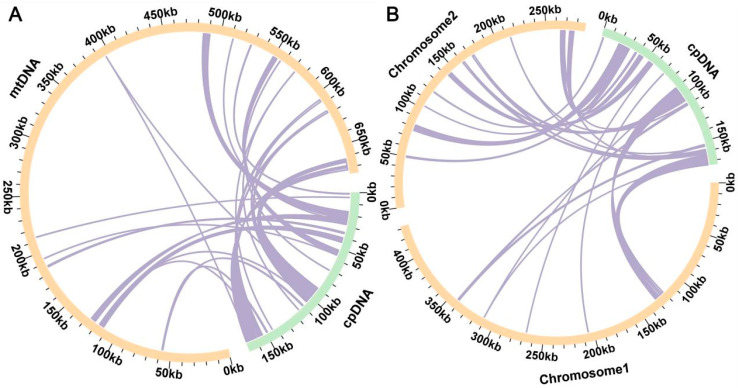
Schematic diagram of gene transfer between chloroplast and mitochondrial genomes in *Z. aethiopica* (**A**) and *Z. odorata* (**B**). The yellow and green arcs represent the mitogenome and chloroplast genomes, respectively, with the purple lines between the arcs corresponding to homologous genomic fragments.

**Figure 5 ijms-24-09566-f005:**
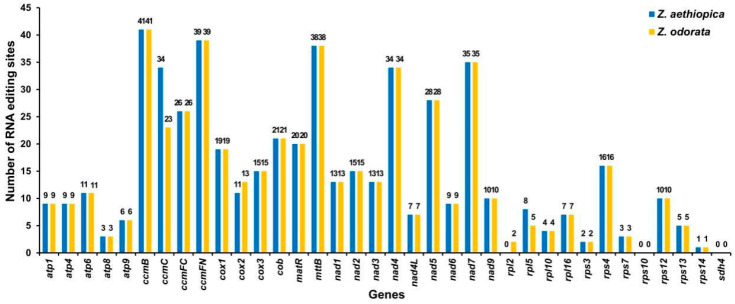
Number of RNA editing sites identified in each PCG of the *Z. aethiopica* and *Z. odorata* mt genomes.

**Figure 6 ijms-24-09566-f006:**
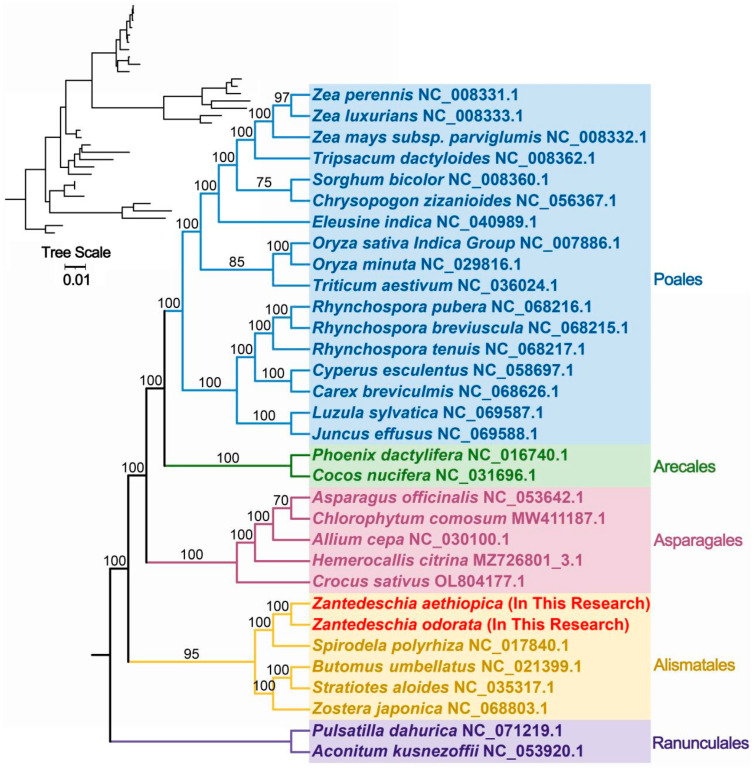
The phylogenetic relationships of *Z. aethiopica* and *Z. odorata* with the 30 other represented land plants. Numbers on each node are bootstrap support values. The colors indicate the families of each species.

**Figure 7 ijms-24-09566-f007:**
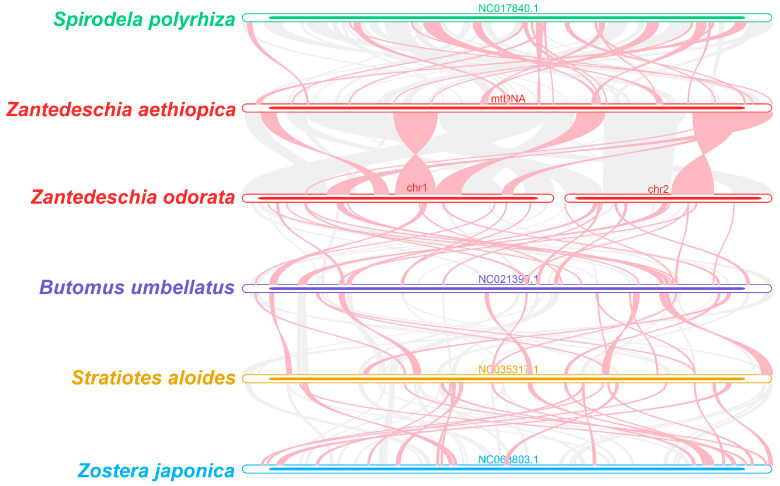
*Z. aethiopica* and *Z. odorata* mitogenomes synteny. Bars indicated the mitogenomes, and the ribbons display the homologous sequences between the adjacent species. The red areas indicate where the reversal occurred, the gray areas indicate regions of good homology. Common blocks less than 0.5 kb in length are not retained, and regions that fail to have a common block indicate that they are peculiar to the species.

**Figure 8 ijms-24-09566-f008:**
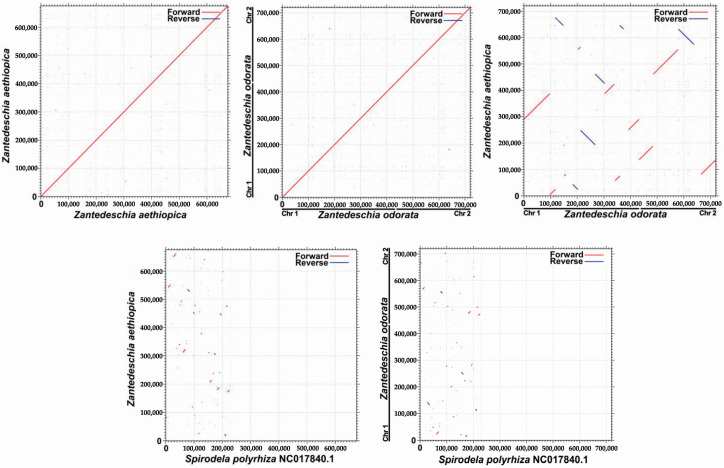
Dot plot analysis of the mitochondrial sequences. The Self dot plots of *Z. aethiopica* and *Z. odorata* are shown in the upper left and middle corner, respectively. The dot plots of *Z. aethiopica* and *Z. odorata* are shown in the upper right corner. The rest are dot plots of *Z. aethiopica* with *S. polyrhiza* and *Z. odorata* with *S. polyrhiza*.

**Figure 9 ijms-24-09566-f009:**
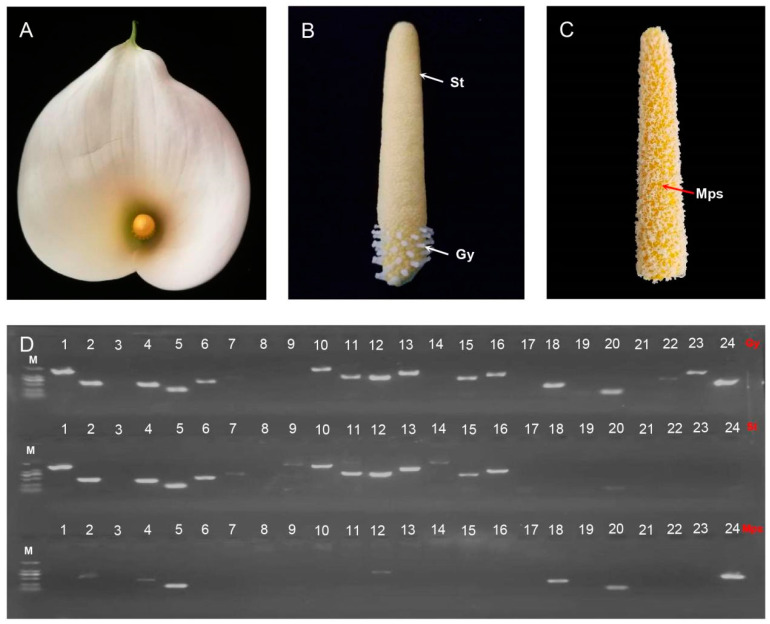
The validation of the 24 core genes in the gynoecium, stamens, and mature pollen grains in *Z. aethiopica* by RT-PCR. (**A**) Spathe of *Z. aethiopica*; (**B**) Gy—gynoecium and St—stamens; (**C**) Mps—mature pollen grains; (**D**) RT-PCR detection of 24 core genes: 1 to 24 represent the *atp1*, *atp4*, *atp6*, *atp8*, *atp9*, *nad1*, *nad2*, *nad3*, *nad4*, *nad4L*, *nad5*, *nad6*, *nad7*, *nad9*, *ccmC*, *ccmFC*, and *ccmFN*, *cox1*, *cox2*, *cox3*, *mttB*, *matR*, and *cob* genes. M: marker (2000 bp, 1000 bp, 750 bp, 500 bp, 250 bp, 100 bp).

**Table 1 ijms-24-09566-t001:** Basic mitochondrial genome information.

Species	Contigs	Type	Length	GC Content
*Z. aethiopica*	Chromosome 1	Circular	675,575 bp	45.85%
*Z. odorata*	Chromosome 1–2	Branched	719,764 bp	45.79%
	Chromosome 1	Circular	432,076 bp	45.95%
	Chromosome 2	Circular	287,688 bp	45.54%

**Table 2 ijms-24-09566-t002:** Genes predicted in the mitogenomes of *Z. aethiopica* and *Z. odoaata*.

	Group of Genes	Name of Genes	
		*Z. aethiopica*	*Z. odorata*
Core genes	ATP synthase	*atp1*, *atp4*, *atp6*, *atp8*, and *atp9*	*Atp1*, *atp4*, *atp6*, *atp8*, and *atp9*
	NADH dehydrogenase	*nad1*, *nad2*, *nad3*, *nad4*, *nad4L*, *nad5*, *nad6*, *nad7*, and *nad9*	*nad1*, *nad2*, *nad3*, *nad4*, *nad4L*, *nad5*, *nad6*, *nad7*, and *nad9*
	Cytochrome c biogenesis	*cob*	*Cob*
	Ubiquinol cytochrome c reductase	*ccmB*, *ccmC*, *ccmFC*, and *ccmFN*	*ccmB*, *ccmC*^2^, *ccmFC*, and *ccmFN*
	Cytochrome c oxidase	*cox1*, *cox2*, and *cox3*	*cox1*, *cox2*, and *cox3*
	Maturases	*matR*	*matR*
	Transport membrane protein	*mttB*	*mttB*
Variable genes	Large subunit of ribosome	*rpl5*, *rpl10*, and *rpl16*	*rpl2*, *rpl5*, *rpl10*, and *rpl16*
	Small subunit of ribosome	*rps3*, *rps4*, *rps7*, *rps12*, *rps13*, and *rps14*	*rps3*, *rps4*, *rps7*, *rps10*, *rps12*, *rps13*, and *rps14*
	Succinate dehydrogenase	*sdh4*	*sdh4*
rRNA genes	Ribosome RNA	*rrn5*, *rrn18*, and *rrn26*	*rrn5*, *rrn18*, and *rrn26*
tRNA genes	Transfer RNA	*trnA*-*UGC*, *trnC*-*GCA*, *trnD*-*GUC*, *trnE*-*UUC*, *trnF*-*GAA*, *trnfM*-*CAU*, *trnG*-*GCC*, *trnH*-*GUG* ^2^, *trnI*-*CAU* ^2^, *trnK*-*UUU*, *trnM*-*CAU* ^2^, *trnN*-*GUU* ^2^, trnP-*UGG*, *trnQ*-*UUG*, *trnS*-*GCU*, *trnS*-*UGA*, *trnV*-*GAC*, *trnW*-*CCA*, and *trnY*-*GUA*	*trnA*-*UGC*, *trnC*-*GCA*, *trnD*-*GUC*, *trnE*-*UUC*, *trnF*-*GAA*, *trnfM*-*CAU*, *trnG*-*GCC*, *trnH*-*GUG* ^2^, *trnI*-*CAU* ^2^, *trnK*-*UUU*, *trnM*-*CAU* ^2^, *trnN*-*GUU*, *trnP*-*UGG*, *trnQ*-*UUG*, *trnS*-*GCU*, *trnS*-*UGA*, *trnV*-*GAC*, *trnW*-*CCA*, and *trnY*-*GUA*

Note: “^2^”: genes with two copies.

## Data Availability

The DNA sequences of two mitogenomes of *Z. aethiopica* and *Z. odorata* are provided as fasta files, along with the annotation information, which is provided as Genbank files and is available through Figshare ((https://doi.org/10.6084/m9.figshare.22591324; accessed on 12 April 2023), (https://doi.org/10.6084/m9.figshare.22591318; accessed on 12 April 2023), (https://doi.org/10.6084/m9.figshare.22591330; accessed on 12 April 2023), and (https://doi.org/10.6084/m9.figshare.22591327; accessed on 12 April 2023)).

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
