# Peer review of "Assembly and Comparative Analysis of the Complete Mitochondrial Genome of Two Species of Calla Lilies (Zantedeschia, Araceae)"

_ijms, 2023, doi:10.3390/ijms24119566_

Round 1

Reviewer 1 Report

Manuscript "Assembly and comparative analysis of the complete mitochondrial genome of two Calla lily" is very interesting.

General comments:
Authors compared the mt genomes of two calla lily species, Zantedeschia aethiopica and Zantedeschia odorata. The Z. aethiopica mt genome was assembled into a single circular chromosome, measuring 675,575 bp in length with a 45.85% GC content.
Authors investigated the core genes in the gynoecium, stamens, and mature pollen grains of the Z. aethiopica mt genome, which revealed maternal mitochondrial inheritance in this species.

Detailed comments:
Comparative analysis between Z. aethiopica and Z. odorata is lacking.

L365: "the mitogenome was visualized using Bandage software" - more interesting is the method used, not the program.

L394-402 - no description of the methods used.

Figure 8: "Dot-plot analysis" - dot-plot is not a form of analysis, but a type of figure

Figure 8: "Dot-plot analysis" - no description of the model and the degree of fit.

Figure 6: "phylogenetic trees" - what method was used to analyze similarity/difference and group objects?

The references are very sloppy.

Paper needs major revision.

Author Response

Dear reviewer,

Thank you very much for your comments and suggestions.

Those comments are all valuable and very helpful for revising and improving our paper, as well as the important guiding significance to our researches. We have studied comments carefully and have made correction which we hope meet with approval. Revised portion are marked in red in the paper. The main corrections in the paper and the responds as flowing:

Point 1: Comparative analysis between Z. aethiopica and Z. odorata is lacking.

Response 1: We have accepted the suggestion and made correction ( We have integrated Table 1 and Table 2 from the original text, the comparative analysis between the Z. aethiopica and Z. odorata mt genomes were reanalyzed).

Point 2: L365: "the mitogenome was visualized using Bandage software" - more interesting is the method used, not the program.

Response 2: We have described the relevant methods in detail.

Point 3: L394-402 - no description of the methods used.

Response 3: We have described the relevant methods in detail. “The hybrid Illumina and Nanopore strategy was used to assemble the mitochondrial genome. First, the Illumina data were used to obtain a graphical mitogenome by GetOrganelle [33]. Second, the GFA format files generated by GetOrganelle were visualized using Bandage software [34], and the individual overlapping groups of chloroplast and nuclear genomes were manually removed to obtain the draft mt genome. Third, we filtered the Illumin and Nanopore data with the draft mt genome as the reference by using samtools [35]. Then, the filtered data were used to conduct a hybrid assembly of the mt genome by using the SPAdes software embedded in Unicycler [36]. Finally, the results were manually corrected, and the final assembled mitochondrial genome was obtained”.

Point 4: Figure 8: "Dot-plot analysis" - dot-plot is not a form of analysis, but a type of figure.

Response 4: We have made corresponding correction.

Point 5: Figure 8: "Dot-plot analysis" - no description of the model and the degree of fit.

Response 5: We have made corresponding correction. “Pairwise comparisons (Z. aethiopica, Z. odorata, and S. polyrhiza) of dot plots were generated, and conserved co-linear blocks were plotted using MAFFT with default parameters [49]”.

Point 6: Figure 6: "phylogenetic trees" - what method was used to analyze similarity/difference and group objects?

Response 6: We have described the relevant methods in detail. “The mitogenomes of closely related species (Table S17) and two outgroups (Pulsatilla dahurica and Aconitum kusnezoffii) were selected and downloaded (https://www.ncbi.nlm.nih.gov) based on their affinity, and then, PhyloSuite (v1.2.3) [42] was used to extract shared genes (18 conserved mitochondrial PCGs, including atp1, atp4, atp6, ccmB, ccmC, ccmFC, ccmFN, cob, cox1, cox2, matR, nad1, nad2, nad4, nad5, nad6, nad7, and nad9). The multiple sequence alignment analysis was conducted using MAFFT [49] with a bootstrap value of 1000. Next, IQ-TREE (v1.6.12) [50] was used for phylogenetic analysis with 5000 ultrafast bootstrapping replicates under the GTR+F+I+I+R2 model. Finally, the evolutionary trees were visualized in ITOL software [51].”

Point 7: The references are very sloppy.

Response 7: We have removed irrelevant references from the entire article, added necessary references, and revised references with incorrect formatting in the article.

Point 8: Paper needs major revision.

Response 8: We have carefully revised the entire article according to the requirements of the editor and other reviewers, and the English editing was conducted through the MDPI language polishing service platform (https://www.mdpi.com/authors/english).

Kind regards,

Guo yanbing

Reviewer 2 Report

The Title is stylistically incorrect.

1.    The Latin names of the family and genus should be added in parentheses after the trivial name of the group under study.

2.    Since there are two species, the genome, respectively, cannot be one.

 I propose the following corrected Title:

Assembly and Comparative Analysis of Complete Mitochondrial Genomes of Two Species of Calla Lilies (Latin Names of Family, Genus)

 Abstract

Lines10-12: The first two sentences are not needed. The general meaning without them will not change. Moreover, the description of the group is in the Introduction.

Lines12-14: Following the style of the Abstract, I propose to replace with “In this study, the mt genomes of two calla species, Zantedeschia aethiopica and Zantedeschia odorata, were assembled and compared for the first time.”

Line 19: cp to mt - The removal of initial part of the text makes it necessary to add the full genomes.

Lines 20-21: remove “precise” and “taxonomic”. Precise is too strong a word for your results. “Taxonomic relationships” is an absurd expression. If relations, then evolutionary.

 Introduction

Space are absent in Lines 38, 41, 45, 48, 55, 65, 71, 72, 74 and 78 (see pdf-ver. with my comments)

Lines 30-31: The beginning of sentence replace with “The representatives…”

Line 33: delete duplicate word “white”

Line 34: “Zatedeschia aethiopica Spreng. and Zantedeschia odorata Perry” place after the words “calla lilies” or even replace calla lilies with two species

Final sentence: The Zantedeschia section includes two species Z. aethiopica Spreng. and Z. odorata Perry characterized by a white spathe and rhizomatous storage organs.

Line 35-38: The same reorganization is required: delete duplicate word “colored” and move species names forward.

Lines 38-41: improve the style, for example, as follows:

The colored species of Zantedeschia are of particular ornamental value due to the prolonged flowering period and vivid flower spathes obtained as a result of interspecific crossing of Z. rehmannii, Z. elliottiana, Z. pentlandii and Z. albomaculata [3,4].

Lines 41-43: re-phrase as follows:

On the contrary, Z. aethiopica has not only decorative, but also practical applications in traditional medicine in some regions of Africa [5].

Line 46: “these two sections” replace with “them”

Line 52: replace beginning of the sentence with “Previous studies have elucidated the PGI mechanism in Zantedeschia and revealed that…”

Line 65-67: delete the unnecessary sentence

Lines 69-72: replace with

Although plant mitogenomes differ significantly in size (from 66kb to 11.3Mb) [13,14], their gene contents remain highly conserved, with 24 core protein-coding genes in angiosperm mitogenomes [add reference!].

Lines 73-75: Common information on the family has wrong placement and could be deleted or moved forward to the most beginning of the Introduction.

Lines 80-82: redundant information should be removed, especially that which is not related to the topic of this study.

I suggest to improve this part of the text as follows:

Due to the complexity of plant mitogenomes study, currently only two Araceae species' mitogenomes have been assembled and deposited in the NCBI: Spirodela polyrhiza (NC_017840) and Amorphophallus albus (NC_066968). The mitogenome sequences is still remain unavailable for most species of the family, impeding the development of new varieties with broader adaptive and commercial traits.

Lines 83-86: Improve style as follows:

In this study, two complete mitochondrial genomes of Zantedeschia species (Z. aethiopica and Z. odorata) were sequenced and compared with available respective data on the other Araceae with aim to better understand the evolution and origin of mitogenome.

 Results Discussion

There are a lot of errors. These sections require extensive English editing.

 Materials and Methods

Notice the spaces between the words. Lots of them are absent.

Lines 354-357: Add geographic coordinates of the collection site(s).

Line 380: Why did you use the old version of MEGA?

Line 393: replace with “Analysis of Phylogeny and Synteny”

Line 394: replace with “mitogenomes”

 In the Conclusions section, you should summarize the most significant findings and their implications, while you list what you did without assessing the fundamental and practical significance of the data obtained.

I suggested some but not all corrections (see my comments to the Authors)

Author Response

Dear reviewer,

Thank you very much for your comments and suggestions.

Those comments are all valuable and very helpful for revising and improving our paper, as well as the important guiding significance to our researches. We have studied comments carefully and have made correction which we hope meet with approval. Revised portion are marked in red in the paper. The main corrections in the paper and the comments are as flowing:

Reviewer comments:

  1. Abstrat

Point 1: The Title is stylistically incorrect.  The Latin names of the family and genus should be added in parentheses after the trivial name of the group under study. Since there are two species, the genome, respectively, cannot be one.  I propose the following corrected Title: Assembly and Comparative Analysis of Complete Mitochondrial Genomes of Two Species of Calla Lilies (Latin Names of Family, Genus)

Response 1:  This was done as requested. “Assembly and Comparative Analysis of the Complete Mitochondrial Genome of Two Species of Calla Lilies (Zantedeschia, Araceae)”

Point 2: Lines10-12: The first two sentences of abstract are not needed. The general meaning without them will not change. Moreover, the description of the group is in the Introduction. 

Response 2: We delete the first two sentences of abstract.

Point 3: Lines12-14: Following the style of the Abstract, I propose to replace with “In this study, the mt genomes of two calla species, Zantedeschia aethiopica and Zantedeschia odorata, were assembled and compared for the first time.”

Response 3: We have accepted the suggestion and made correction.

Point 4Line 19: cp to mt - The removal of initial part of the text makes it necessary to add the full genomes.

Response 4: This was done as requested. “Analyses of codon usage, sequence repeats, gene migration from chloroplast to mitochondrial, and RNA editing were conducted for both Z. aethiopica and Z. odorata mt genomes.”

Point 5Lines 20-21: remove “precise” and “taxonomic”. Precise is too strong a word for your results. “Taxonomic relationships” is an absurd expression. If relations, then evolutionary.

Response 5: We have accepted the suggestion and made correction.

  1. Introduction

Point 1: Space are absent in Lines 38, 41, 45, 48, 55, 65, 71, 72, 74 and 78 (see pdf-ver. with my comments)

Response 1: We have made corresponding correction.

Point  2: Lines 30-31: The beginning of sentence replace with “The representatives…”

Response 2: We have made unified correction based on the 12th modification suggestion (Lines 73-75: Common information on the family has wrong placement and could be deleted or moved forward to the most beginning of the Introduction) as fllower: “The Araceae family, one of the largest monocot families, comprises approximately 3,667 species across 143 genera [1]. Many species within the family hold global significance in medicine, horticulture, and as edible plants. The representatives genus Zantedeschia within……”

Point 3: Line 33: delete duplicate word “white”

Response 3: We delete duplicate word “white”.

Point 4: Line 34: “Zatedeschia aethiopica Spreng. and Zantedeschia odorata Perry.” place after the words “calla lilies” or even replace calla lilies with two species. Final sentence: “The Zantedeschia section includes two species Z. aethiopica Spreng. and Z. odorata Perry. characterized by a white spathe and rhizomatous storage organs”.

Response 4: We accept this suggestion.

Point 5: Line 35-38: The same reorganization is required: delete duplicate word “colored” and move species names forward.

Response 5: This was done as requested. “In contrast, the Aestivea section comprises     Z. albomaculata, Z. elliottiana, Z. jucunda, Z. pentlandii, Z. rehmannii, and Z. valida with a various spathe colors and tuberous storage organs [2,3]”.

Point 6: Lines 38-41: improve the style, for example, as follows: “The colored species of Zantedeschia are of particular ornamental value due to the prolonged flowering period and vivid flower spathes obtained as a result of interspecific crossing of Z. rehmannii, Z. elliottiana, Z. pentlandii and Z. albomaculata [3,4]”.

Response 6: We have accepted the suggestion and made correction.

Point 7: Lines 41-43: re-phrase as follows: “On the contrary, Z. aethiopica has not only decorative, but also practical applications in traditional medicine in some regions of Africa [5]”.

Response 7: We have accepted the suggestion and made correction.

Point 8: Line 46: “these two sections” replace with “them”

Response 8: We have accepted the suggestion and made correction. 

Point 9: Line 52: replace beginning of the sentence with “Previous studies have elucidated the PGI mechanism in Zantedeschia and revealed that…”

Response 9: We have accepted the suggestion and made correction.

Point 10: Line 65-67: delete the unnecessary sentence

Response 10: We have deleted the unnecessary sentence: “The rapid advancement of sequencing technology has facilitated the reporting of an increasing number of plants mt genomes”.

Point 11: Lines 69-72: replace with “Although plant mitogenomes differ significantly in size (from 66kb to 11.3Mb) [13,14], their gene contents remain highly conserved, with 24 core protein-coding genes in angiosperm mitogenomes [add reference!].”

Response 11: We have replaced the sentence “Although plant mitogenomes differ significantly in size, 69 their gene contents remain highly conserved, with 24 core protein-coding genes in angiosperm mitogenomes and mitochondrial lengths ranging from a minimum of 66kb[13] to a maximum of 11.3Mb[14]. ” with “Although plant mitogenomes differ significantly in size (from 66kb to 11.3Mb) [16,17], their gene contents remain highly conserved, with 24 core protein-coding genes in angiosperm mitogenomes [18].”

Point 12: Lines 73-75: Common information on the family has wrong placement and could be deleted or moved forward to the most beginning of the Introduction.

Response 12: We have made unified correction based on the 2th modification suggestion (Lines 30-31: The beginning of sentence replace with “The representatives…”)

Point 13: Lines 80-82: redundant information should be removed, especially that which is not related to the topic of this study. I suggest to improve this part of the text as follows: Due to the complexity of plant mitogenomes study, currently only two Araceae species' mitogenomes have been assembled and deposited in the NCBI: Spirodela polyrhiza (NC_017840) and Amorphophallus albus (NC_066968). The mitogenome sequences is still remain unavailable for most species of the family, impeding the development of new varieties with broader adaptive and commercial traits.

Response 13: We have removed redundant information and accepted the text suggestion of this part.

Point 14: Lines 83-86: Improve style as follows: In this study, two complete mitochondrial genomes of Zantedeschia species (Z. aethiopica and Z. odorata) were sequenced and compared with available respective data on the other Araceae with aim to better understand the evolution and origin of mitogenome.

Response14: We have accepted the suggestion and made correction.

  1. Results Discussion

Point 1: There are a lot of errors. These sections require extensive English editing.

Response 1: Our manuscript have conducted undergo extensive English revisions by  the editing services (https://www.mdpi.com/authors/english).

  1. Materials and Methods

Point 1: Notice the spaces between the words. Lots of them are absent.

Response 1: We have accepted the suggestion and made correction. Then, we have also revised the other errors in this section.

Point 2: Lines 354-357: Add geographic coordinates of the collection site(s).

Response 2: We have added geographic coordinates of the collection site(s).

Point 3: Line 380: Why did you use the old version of MEGA?

Response 3: We have analyzed using the latest version of MEGA and obtained the same results.

Point 4: Line 393: replace with “Analysis of Phylogeny and Synteny”

Response 4: We have accepted the suggestion and made correction.

Point 5: Line 394: replace with “mitogenomes”

Response 5: We have accepted the suggestion and made correction.

  1. Conclusions

Point 1: In the Conclusions section, you should summarize the most significant findings and their implications, while you list what you did without assessing the fundamental and practical significance of the data obtained.

Response 1: We have accepted the suggestion and made correction. Conclusions: 

In this study, the two mt genomes of calla lilies species (Z. aethiopica and Z. odorata) were sequenced, assembled, and annotated, and comprehensive analyses were conducted based on the DNA and amino acid sequences of the annotated genes. The two calla species’ mitogenomes shared similar gene contents but varied in structure. The Z. aethiopica mt genome length of the circular structure was 675,575 bp and the GC content was 45.85%, while the Z. odorata mt genome consists of two circular chromosome structures with a total genome size of 719,764 bp and a GC content of 45.79%. By comparing the mt and cp genomes, 29 and 25 homologous fragments were found in the two species, respectively. Furthermore, we analyzed codon usage, repeated sequences, RNA editing, and synteny. Moreover, phylogenetic trees based on the mitogenomes of 30 species contributed to the taxonomic classification of Z. aethiopica and Z. odorata. Lastly, we demonstrated maternal mitochondrial inheritance in Z. aethiopica using RT-PCR. This study provides a pivotal genomic resource for understanding and utilizing this vital ornamental plant in the future, particularly in elucidating the PGI mechanism in plants.

Kind regrads,

Guo yanbing

Round 2

Reviewer 1 Report

The authors supplemented the manuscript with a list of the programs used. Unfortunately, there is still no information on what method was used to analyze similarity/differences and what method was used to perform clustering of objects!

Paper needs major revision.

Author Response

Dear reviewer,

Thank you very much for your comments and suggestions in the second round of review.

Those comments are all valuable and very helpful for revising and improving our paper, as well as the important guiding significance to our researches. We have revised the manuscript, according to the comments and suggestions of reviewers and editor, and responded, point by point to, the comments as listed below. Revised portion are marked in red in the paper. The main corrections in the paper and the responds as flowing: 

Point 1: The authors supplemented the manuscript with a list of the programs used. Unfortunately, there is still no information on what method was used to analyze similarity/differences and what method was used to perform clustering of objects!

Response 1: This was done as requested. We have rewritten the relevant methods in detail.

(1)The assembly of itochondrial genomes:“We used Flye (v.2.9.1-b1780) [38] to de novo assemble the long-reads of the two Zantedeschia species with the parameters of ‘--min-overlap 4,000’. For the assembled contigs, the draft mitogenome based on Nanopore long-reads was identified using BLASTn [39] Specifically, here, we used makeblastdb to construct a database for the assembled sequences by Flye, and then we used the conserved mitochondrial genes from Liriodendron tulipifera (NC_021152.1) as the query sequence to identify contigs containing conserved mitochondrial genes. We then mapped the short-reads and long-reads to these contigs, and all mapped reads were remained using BWA and SAMTools [40, 41]. Finally, we combined Illumina short-reads and Nanopore long-reads for hybrid assembly by using Unicycler (v0.5.0) [42], with the parameters of ‘--kmers 27, 53, 71, 87, 99, 111, 119, 127’. In this step, the mapped Illumina short-reads before were assembled by calling spades [43], and then, the Nanopore long-reads were used to resolve the regions of repetitive sequences of the assembly by calling minimap2 [44]. The GFA format files generated by Unicycler were visualized using Bandage [45]. Unicycler generated one circular contigs for Z. aethiopica and two circular contigs for Z. odorata.”

(2)The construction of phylogenetic trees: The mitogenome of closely related species were selected and downloaded (https://www.ncbi.nlm.nih.gov) based on their affinity. We downloaded a total of 30 mitogenomes, including two outgroups (Pulsatilla dahurica: NC_071219.1 and Aconitum kusnezoffii: NC_053920.1). These species (Table S17) were then used for phylogenetic analysis along with the two Zantedeschiaspecies we newly sequenced. Firstly, the 24 core orthologous genes among the analyzed species were extracted by using PhyloSuite (v.1.2.2) [50]. Then, the corresponding nucleotide sequences were aligned using MAFFT (v7.471) [57]. Next, these aligned sequences were concatenated and used to construct the phylogenetic trees. The maximum likelihood (ML) method was implemented in IQ-TREE (version 2.1.4-beta) [58] under the GTR+F+I+I+R2 model. The bootstrap analysis was performed with 1,000 replicates. The phylogenetic tree was edited on the online website ITOL [59].

(3)Analysis of synteny: “Four closed species (Spirodela polyrhiza (NC_017840.1), Butomus umbellatus (NC_021399.1), Zostera japonica (NC_068803.1), and Stratiotes aloides (NC_035317.1)) were selected for the co-linear analysis with aethiopica and Z. odorata. Conserved homologous sequences, referred to as collinear blocks, were identified using the BLASTn program [39] with parameters of ‘-evalue 1e-5, -word_size 9, -gapopen 5, -gapextend 2, -reward 2, and -penalty -3’. Only collinear blocks that exceeded 500 bp in length were chosen for further analysis. MCscanX [60] was utilized to generate the multiple synteny plot based on the results obtained from the BLASTn program. Pairwise comparisons (Z. aethiopica, Z. odorata, and S. polyrhiza) of dot plots were generated, and conserved co-linear blocks were plotted using MAFFT with default parameters [57]. ”

Thank you again for you pay for our manuscript. We look forward to hearing from you soon.

Kind regards,

Guo yanbing

Round 3

Reviewer 1 Report

Authors have incorporated all the suggestions, accordingly. I recommend this article to publish in current version.